# Comparisons between the WRF data assimilation and the GNSS tomography technique in retrieving 3D wet refractivity field in Hong Kong

**Zhaohui Xiong [1], Bao Zhang [1, *] and Yibin Yao [1]**

[1]  School of Geodesy and Geomatics, Wuhan University, Wuhan 430079;

**\*** Correspondence: sggzb@whu.edu.cn; Tel.: +86-027-6875-8403

**Abstract:** Water vapor plays an important role in various scales of weather processes. However, there are limited means to accurately describe its 3-dimensional (3D) dynamical changes. The data assimilation technique and the Global Navigation Satellite System (GNSS) tomography technique are two of the limited means. Here, we conduct an interesting comparison between the GNSS tomography technique and the Weather Research and Forecasting Data Assimilation (WRFDA) (a representative of the data assimilation models) in retrieving Wet Refractivity (WR) in Hong Kong area during a wet period and a dry period. The GNSS tomography technique is used to retrieve WR from the GNSS slant wet delay. The WRFDA is used to assimilate the zenith tropospheric delay to improve the background data. The radiosonde data are used to validate the WR derived from the GNSS tomography, the WRFDA output, and the background data. The Root Mean Square (RMS) of the WR derived from the tomography results, the WRFDA output, and the background data are 6.50 mm/km, 4.31 mm/km, and 4.15 mm/km in the wet period. The RMS becomes 7.02 mm/km, 7.26 mm/km, and 6.35 mm/km in the dry period. The lower accuracy in the dry period is mainy due to the sharp variation of WR in the vertical direction. The results also show that assimilating GNSS ZTD into the WRFDA only slightly improves the accuracy of the WR and that the WRFDA WR is better than the tomographic WR in most cases. However, in a special experimental period when the water vapor is highly concentrated in the lower troposphere, the tomographic WR outperforms the WRFDA WR in the lower troposphere. When we assimilate the tomographic WR in the lower troposphere into the WRFDA, the retrieved WR is improved.

**Keywords:** GNSS Tomography; Wet Refractivity; Weather Research and Forecasting model; Data Assimilation

## 1 Introduction

Water vapor (WV), mostly contained in the troposphere, plays an important role in various scales of atmospheric processes. But due to its active nature, there are limited models and techniques that can accurately describe or monitor its 3-dimensional (3D) dynamical changes (Rocken et al., 1993).

The development of Global Navigation Satellite System (GNSS) technique and the densely deployed GNSS receivers provide us the opportunity to monitor the WV field in near real time. When GNSS signal travels through the neutral atmosphere, it undergoes time delay and bending due to atmospheric refractivity. This effect is usually called the tropospheric delay in the GNSS community (Altshuler, 2002). The tropospheric delay is usually considered as the product of the zenith delay and the mapping function (Lanyi, 1984; Niell, 1996). The Zenith Tropospheric Delay (ZTD) consists of two parts: the hydrostatic part and the wet part. The wet delay is mainly associated with the WV and reflects WV content in the troposphere. Bevis et al. (1992) introduced the principle of using GNSS Zenith Wet Delay (ZWD) to retrieve the Precipitable Water Vapor (PWV). Since then, many scientists carried out the GNSS PWV experiments (Askne and Nordius,1987; Bokoye et al., 2003; Yao et al., 2014; Lu et al., 2015; Shoji and Sato, 2016). Now, the GNSS PWV can be retrieved with an uncertainty of 1-2 mm in post-processing (Tregoning et al., 1998; Adams et al., 2011; Grejner-Brzezinska, 2013) or real-time modes (Yuan et al., 2014; Li et al., 2014; Li et al., 2015).

The GNSS WV tomography technique was first proposed to monitor the 3D or 4D WV in 2000 (Flores et al., 2000; Seko et al., 2000; Hirahara et al., 2000). Since then, many scientists have proposed refined methods to improve the GNSS WV tomography (Flores et al., 2001; Nilsson and Gradinarsky, 2006; Rohm and Bosy, 2011; Wang et al., 2014; Wang et al., 2014; Zhao and Yao, 2017). The tomographic inversion algorithm can be roughly categorized into two groups. One group solves the tomography equation in the least squares scheme or in the Kalman filter scheme with additional constraints or using a priori information (Flores et al., 2000; Rohm and Bosy, 2011; Cao et al., 2006; Zhang et al., 2017). The other group uses the algebraic reconstruction algorithm or similar methods (Bender et al., 2011; Wang et al., 2014; Zhao and Yao, 2017). Some scientists also use different methods from the above to solve the GNSS WR tomography problem (Nilsson and Gradinarsky, 2006; Perler et al., 2011; Altuntac, 2015). Besides the algorithm improvement, some scientists tried to optimize the voxel division (Chen and Liu, 2014) or use virtual reference stations (Adavi and Mashhadi-Hossainali, 2014) or use additional GNSS rays (Zhao and Yao, 2017) to increase the effective GNSS rays and thus improve the tomography results. Though the tomography technique has the advantages of (1) free of weather conditions and (2) retrieve 3D WR field in near real time, it still suffers some problems. The sparse distribution of the GNSS receivers and the bad satellite-receiver geometry lead to serious ill-posed and ill-conditioned problems, and also limit the WR retrieve resolution in both vertical and horizontal domains.

Besides the GNSS tomography technique, the WR can also be retrieved by data assimilation which is based on numerical weather prediction (NWP) models (Perler et al., 2011). The Weather Research and Forecasting (WRF) model is a state-of-the-art atmospheric modeling system that is used to simulate the dynamic processes of the atmosphere (Jankov et al., 2005; Carvalhoaabc et al., 2012). It is mainly developed and supported by Mesoscale and Microscale Meteorology (MMM) Laboratory of the National Center for Atmospheric Research (NCAR). And the WRF Data Assimilation (WRFDA) is designed to obtain the best estimate of the actual atmospheric state at any analysis time (Barker et al., 2004; Huang et al., 2008; Barker et al., 2012; Singh et al., 2017). Many studies have demonstrated that assimilating ZTD/PWV into WRFDA can improve the reanalysis water vapor field (Pacione et al., 2001; Faccani et al., 2005; Boniface et al., 2012; Bennitt and Jupp, 2012; Moeller et al., 2016; Lindskog et al., 2017). Besides the WRFDA model, the Japan Meteorological Agency (JMA) Mesoscale Numerical Weather Prediction Model (Nakamura et al., 2004) and AROME NWP system (Boniface et al., 2009) can also make use of ZTD\PWV data assimilation.

Though the GNSS tomography technique and the data assimilation technique belong to different fields, both of them could retrieve 3D WR field. It will be interesting to compare their capabilities in retrieving WR field under different weather conditions and to explore the feasibility to combine them. Such results may provides insights for the NWP community about this new technique and the possibility of assimilating the tomography results into the NWP models. For the GNSS community, they will get a better understanding of the WRF data assimilation and its capability in simulating the water vapor field. For this purpose, we conduct GNSS tomography and data assimilation experiments in Hong Kong area using SatRef Network in a wet period and a dry period. WR fields retrieved from GNSS tomography and WRFDA outputs are validated by the radiosonde data. We also explore the feasibility of assimilating the GNSS tomographic WR into the WRFDA to further improve the WR field.

## 2 Research Area and Data Analysis

The study area is within 113.75°E-114.5°E and 22°N-22.6°N as shown in Figure 1. There are 15 continues GNSS stations belonging to the Hong Kong SatRef Network deployed in the study area. They are all equipped with Leica GNSS receivers and antennas to receive the GNSS signals and automatic meteorological devices to record the temperature, pressure, and relative humidity. The average inter-distance between stations is about 10 km. The altitudes of the highest station (HKNP) and the lowest station (HKLM) are 354 m and 10 m. In GNSS tomography, we regard a network whose altitude differences of its stations are less than 1 km as a flat network. Therefore, the SatRef network is a flat network.

Two periods of GNSS observation data are processed to generate ZTD and Slant Wet Delay (SWD). One is a wet period from July 20 to 26, 2015 when Hong Kong suffered the heaviest daily rainfall of 2015 (191.3 mm rainfall on July 22). The other is a dry period from August 1 to 7, 2015 when Hong Kong is rainless. The details about the GNSS data processing and the SWD reconstruction can be found in Applendix A.

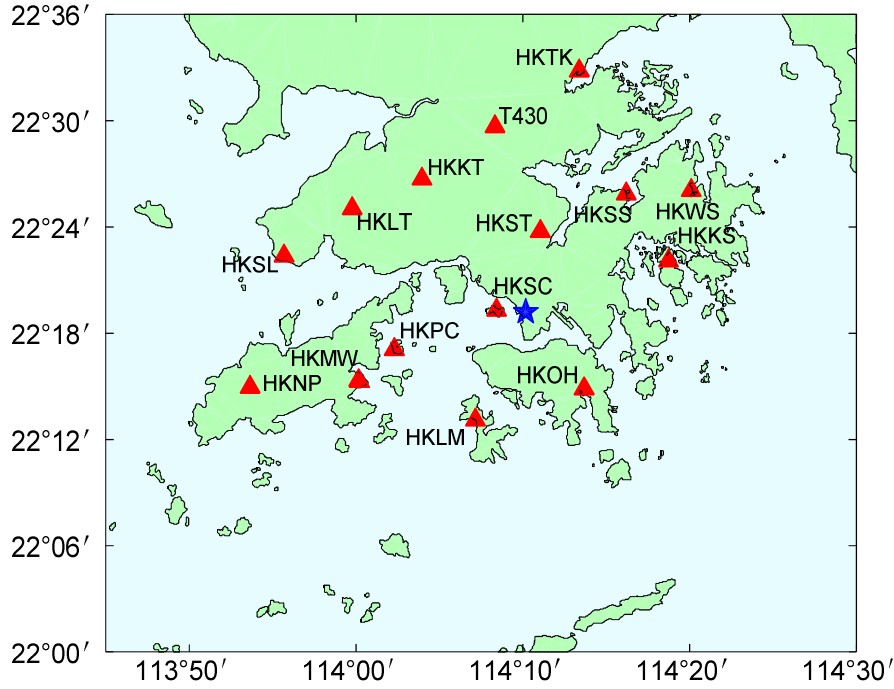

**Figure 1.** Research area of the experiment. The red triangles indicate the GNSS stations and the blue star indicates the radiosonde station in Hong Kong.

## 3 Method

### *3.1. WRF Data assimilation*

The WRF model version 3.7 is used in this study. The WRFDA-3DVAR is used to assimilate the GNSS ZTD to improve the background data. The horizontal resolution of WRFDA output is set to 3 km. And the atmosphere is vertically divided into 45 layers. The pressure of the top layer is 50 hpa. There are 10 layers in the planetary boundary layer (PBL). We use the ZTD error output by the Bernese 5.0 software as obervation error. We use the reanalysis data from European Center for Medium-Range Weather Forecasts (ECMWF) ERA-Interim pressure levels and surface data as the background data, whose spatial resolution is $0.75° \times 0.75°$. And we run the WRFDA model at 0:00 UTC and 12:00 UTC, corresponding to the radiosonde observation time. The procedures to do the assimilation experiments are shown in Figure 2.

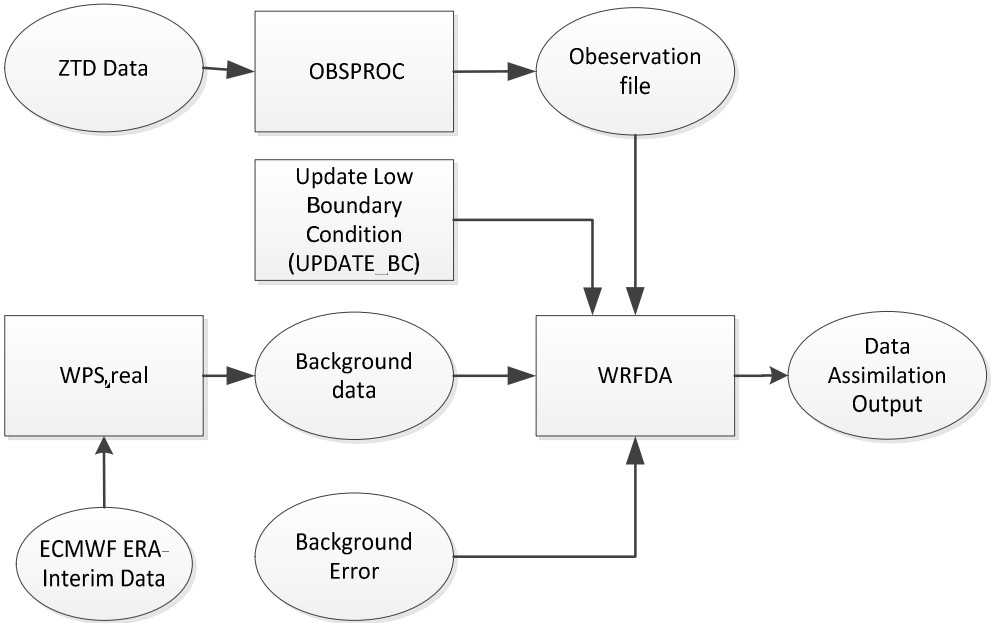


**Figure 2.** Flowchart of data assimilation using the WRF model.
The background data are processed by WRF preprocessing system (WPS). The WRFDA is run with the generic
CV3 option, and the default background error is adopted in this study. The GNSS ZTDs are the input
observations for WRFDA. We run WRFDA to obtain the data assimilation output, labeled as Output1. The
output from WPS and real.exe is labeled as Output2. We compare the WR derived from Output1 and Output2.
We use Equation (1) to calculate WR (Vedel and Huang, 2004) from the Output1 and Output2.

$$WR = \frac{P_w}{T} \times (k_1 + \frac{k_2}{T}) \tag{1}$$

where $P_w$ is the water vapor pressure in each grid point in Pascal, $T$ is the temperature in each grid point in
Kelvin. $k_1 = 2.21 \times 10^{-7}$ K/Pa, $k_2 = 3.73 \times 10^{-3}$ K²/Pa. We use Equation (2) to calculate $P_w$.

$$P_w = \frac{p \times q}{0.622} \tag{2}$$

where $p$ is the pressure in Pascal, $q$ is the specific humidity in g/g.
The WRFDA has many options for different physical parameterizations. In order to find the best choice for the
data assimilation experiment, we follow Chien et al. (2006) to set 12 schemes to do the sensitivity tests, which
are listed in Table 1. We carry out the sensitivity test at 00:00 UTC 22nd July in 2015. The domain size is set to
$30 \times 24$ grids which just cover the study area. The grid size is 3 km $\times$ 3 km. We run WRFDA using the different
setting schemes. The radiosonde data are used to validate the wet refractivity derived by the WRFDA output.
Table 1 shows that all schemes have the same bias, standard deviation (STD), and Root Mean Square (RMS),
which suggests that the output wet refractivity is not affected by the physical parameterization settings in
WRFDA.
In this study, we use the Kain-Fritsch scheme (Kain and Frisch, 1990), WRF Single-Moment (WSM) 5-class
scheme (Hong et al., 2004) and Yonsei University PBL scheme (Hong et al., 2006), which are the same to Chien
et al. (2006). The other physical options include unified Noah land-surface model (Tewari et al., 2004), Revised
MM5 Monin-Obukhov scheme (Monin and Obukhov, 1954). The Rapid Radiative Transfer Model (Mlawer et
al., 1997) and Dudhia's scheme (Dudhia, 1989) are used for longwave radiation and shortwave radiation,
respectively.

**Table 1.** Physical parameterization schemes and statistics of bias, RMS, and STD of retrieved WR using different schemes. Unit is mm/km.

|  | PBL physics | cumulus physics | microphysics | bias | STD | RMS |
|---|---|---|---|---|---|---|
| 1 | Yonsei University | Kain-Fritsch | WSM 5-class | -3.95 | 6.55 | 7.51 |
| 2 | Yonsei University | Betts-Miller-Janjic | WSM 5-class | -3.95 | 6.55 | 7.51 |
| 3 | Yonsei University | Grell-Freitas ensemble | WSM 5-class | -3.95 | 6.55 | 7.51 |
| 4 | Yonsei University | Kain-Fritsch | Ferrier | -3.95 | 6.55 | 7.51 |
| 5 | Yonsei University | Betts-Miller-Janjic | Ferrier | -3.95 | 6.55 | 7.51 |
| 6 | Yonsei University | Grell-Freitas ensemble | Ferrier | -3.95 | 6.55 | 7.51 |
| 7 | Mellor-Yamada-Janjic | Kain-Fritsch | WSM 5-class | -3.95 | 6.55 | 7.51 |
| 8 | Mellor-Yamada-Janjic | Betts-Miller-Janjic | WSM 5-class | -3.95 | 6.55 | 7.51 |
| 9 | Mellor-Yamada-Janjic | Grell-Freitas ensemble | WSM 5-class | -3.95 | 6.55 | 7.51 |
| 10 | Mellor-Yamada-Janjic | Kain-Fritsch | Ferrier | -3.95 | 6.55 | 7.51 |
| 11 | Mellor-Yamada-Janjic | Betts-Miller-Janjic | Ferrier | -3.95 | 6.55 | 7.51 |
| 12 | Mellor-Yamada-Janjic | Grell-Freitas ensemble | Ferrier | -3.95 | 6.55 | 7.51 |

In order to figure out how sensitive the wet refractivity output is to the domain size, we carry out another
sensitive test at 00:00 UTC 22nd July in 2015. And we increase the domain size gradually from 30 × 24 grids to
190 × 184 grids. In each run, we validate the wet refractivity derived by the WRFDA output using the radiosonde
data. The statistical results of the sensitivity tests are shown in Figure 3. It shows that the smaller domain size
has the smaller bias, STD, and RMS. So, the domain size of the data assimilation experiment is set to 30 × 24
grids which just cover the study area.

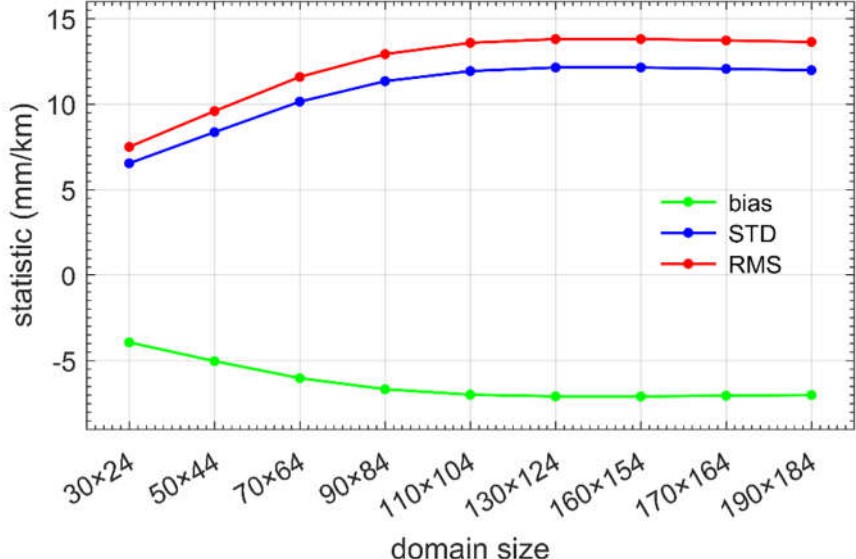


**Figure 3.** WR Statistics of sensitive test of domain size validated by radiosonde data.

*3.2. GNSS tomography*
The limited number of stations, flat vertical distribution of stations, and bad satellite-station geometry impose
serious ill-posed problem in the WR tomography. To well handle this problem, we use the tomography method
proposed by Zhang et al. (2017). This method is based on the adaptive Laplacian smoothing and Helmert
Variance Component Estimation. It also uses the meteorological data from each GNSS station to constrain the
WR near the ground. This tomography strategy is free of a priori information, which makes it an independent
technique and thus ensures the fairness when the tomography technique is compared with the WRFDA model.
The WR can be retrieved directly by this tomography strategy when the SWDs are used as observations. The
troposphere is vertically divided into 13 layers with a constant thickness of 800 meters, and horizontally divided
into grids whose resolution is ~10 km in longitudinal direction and ~8 km in latitudinal direction. The
tomography algorithm is described as follows:

$$\left. \begin{array}{l} \mathbf{Y} = \mathbf{AX} \\ \mathbf{0} = \mathbf{VX} \\ \mathbf{0} = \mathbf{HX} \\ \mathbf{0} = \mathbf{BX} \\ \mathbf{X}_m = \mathbf{X} \end{array} \right\} \tag{3}$$


where the first equation is the observation equation, $\mathbf{Y}$ is the vector of SWDs, $\mathbf{A}$ is the design matrix
consisting of intercepts in each voxel, $\mathbf{X}$ is the vector of WR in each voxel. The second to the forth
equations in Equation (3) are the vertical, horizontal, and boundary constraints. The fifth equation is used
to constrain the WR near the ground using the meteorological data at each GNSS station. $\mathbf{V}$, $\mathbf{H}$, and
$\mathbf{B}$ are design matrix for constraint equations. The boundary constraints are established by setting the
WR in the top layer to 0. The vertical and horizontal constraints are established by Laplacian smoothing
in the vertical and horizontal directions, respectively. The Laplacian smoothing can be described as:

$$x_1 + x_2 + x_3 + x_4 - q x_0 = 0 \tag{4}$$


where the WR $x_0$ equals the weighted average WR of its nearest four voxels in the same plane, $q$ is
the smoothing factor.
In a least square scheme, the solution can be found by:

$$\mathbf{X} = (\mathbf{A}^{\mathrm{T}}\mathbf{A} + \lambda_1 \mathbf{V}^{\mathrm{T}}\mathbf{V} + \lambda_2 \mathbf{H}^{\mathrm{T}}\mathbf{H} + \lambda_3 \mathbf{B}^{\mathrm{T}}\mathbf{B} + \lambda_4)^{-1}(\mathbf{A}^{\mathrm{T}}\mathbf{Y} + \lambda_4 \mathbf{X}_m) \tag{5}$$


Where $\lambda_i$ (i = 1, 2, 3, 4) are the weights of corresponding constraints.
In Zhang et al. (2017), the solution is found in an iterative feedback-update process, which is be simply
described as follows:
(a) Establish the initial constraints and initialize their weights as 1, namely $\lambda_1 = \lambda_2 = 1$, $\lambda_3$ is set to a large
value, $\lambda_4$ is set to 1; $\lambda_3$ and $\lambda_4$ are not updated in the following run.
(b) Determine the values of $\lambda_1$ and $\lambda_2$ by Helmert Variance Component Estimation method and calculate
the tomography solutions by Equation (5);
(c) Update the smoothing factors by using the solutions in (b):

$$q = \begin{cases} n & \text{if} \quad x_0 < x_m \\ \dfrac{\sum\limits_{i=1}^{n} x_i}{x_0} & \text{if} \quad x_0 > x_m \end{cases} \tag{6}$$

where $n$ is the number of voxels used to calculate the weighted average. $x_m$ is a threshold set to prevent
updating the smoothing factor by inaccurate solutions. The initial value for $x_m$ is half of the maximum wet
refractivity in the solutions. $x_m$ is updated by multiplying $x_m$ by a scale factor, say 0.9, after each run until it
is no larger than 3 times the mean square error of $\mathbf{X}$.
(d) Use the new smoothing factors in (c) to update the horizontal and vertical constraints and redo (b) and (c)
until the mean square error of the solution differences between this run and the previous run approaches a stable
value. In practice, we set a threshold of 20 iterations which is enough to ensure a stable solution.
**4 Results**
The radiosonde data are used to validate the WR derived from GNSS tomography, the Output1 and the Output2.
Since the radiosonde launches at 0:00 and 12:00 UTC daily, the WR at these epochs are validated. Equation (1)
is also used to calculate WR from radiosonde data. The vertical coordinates of the Output1 and the Output2 are
converted to geopotential heights by NCAR Command Language (NCL) (UCAR/NCAR/CISL/VETS, 2013)
and the geodetic heights of tomographic results are converted to normal height. The slight differences between
geopotential heights and normal heights are neglected. We interpolate the Radiosonde to tomographic nodes
since the former has a much higher resolution ~23 layers from 0 to 10 km height than the latter (13 layers) and
thus we can get a higher interpolation accuracy. We use a bilinear interpolation method in the horizontal domain
and a linear interpolation method in the vertical direction. By these methods, we interpolate the WR derived
from the Output1, the Output2 and radiosonde data to the tomography nodes. Finally, the WR are validated by
the radiosonde data. For simplicity, WR from radiosonde data, and GNSS tomography are denoted as
"Radiosonde", and "Tomography" hereinafter.
Figures 4 and 5 show the vertical profiles of the Radiosonde, the Output1, the Output2 and the Tomography in
the July and August periods, respectively. The Output1, the Output2, and the Tomography agree well with the
Radiosonde, which indicates that these three methods successfully retrieved the vertical profile of the WR. It is
also observed that the Output1, the Output2, and the Tomography agree better with the Radiosonde in the July
period than in the August period. This difference should be due to the vertical distribution of WR. Though Hong
Kong suffered heavy rain in the July period, the WR was more evenly distributed from 0 to 10 km height than
that in the August period. In the dry August period, the WR was highly concentrated in the lower troposphere
(< 6 km) and its vertical changes were very sharp. This situation decreased the performance of the tomography
technique and the data assimilation technique. This also indicate that the tomography technique has decreased
capabilities in retrieving WR in highly changing troposphere. Compared with the Output2, the Output1 is
slightly improved by reducing the mean absolute error (MAE) by 1.25 mm/km. The difference between the
Tomography and the Output1 is obvious at some time epochs in the dry period (e.g., 12:00 on August 4 and 5).

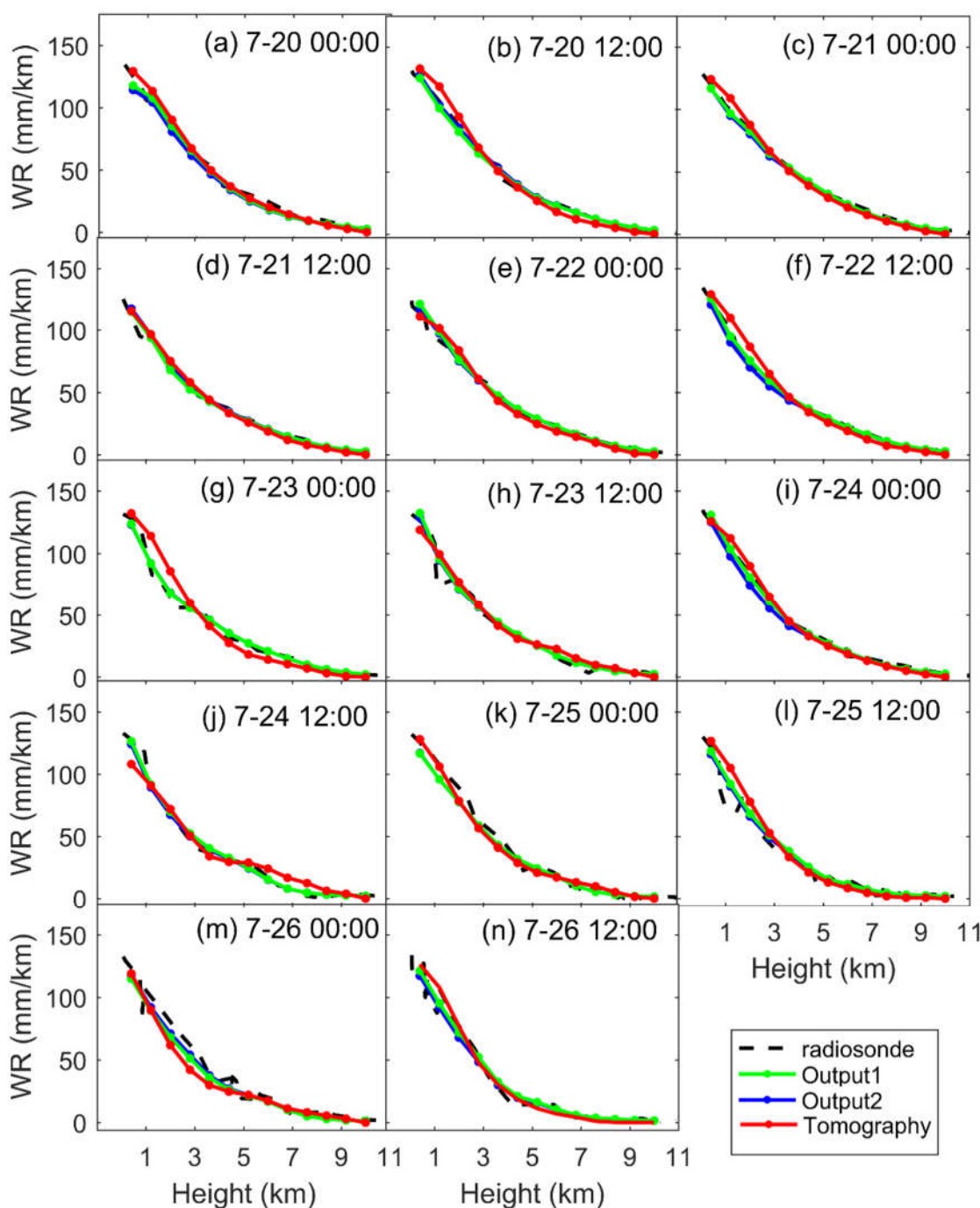


**Figure 4.** Comparisons among WR derived from Output1, Output2, Tomography, and Radiosonde in the wet period, 2015.

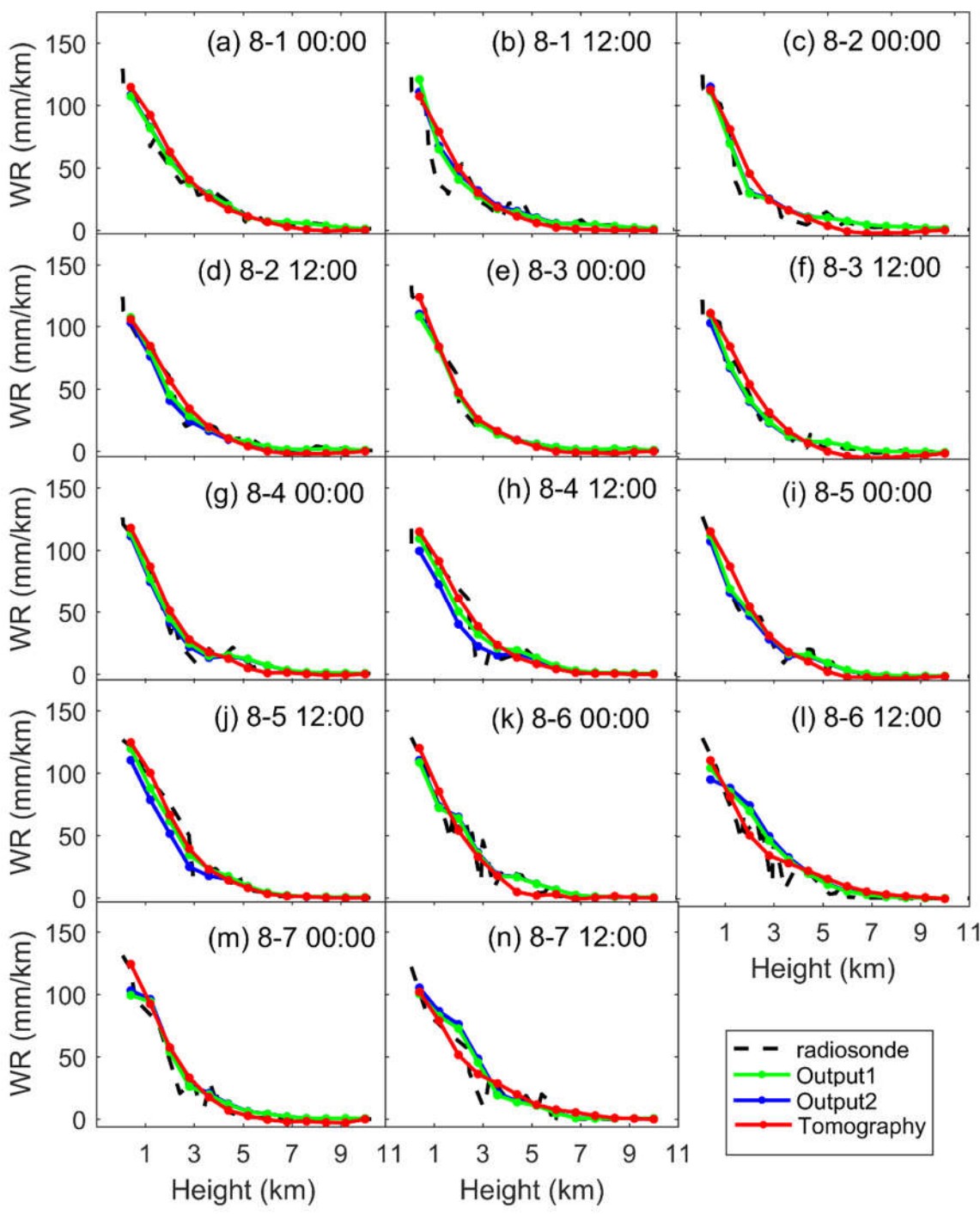


**Figure 5.** Comparisons among WR derived from Output1, Output2, Tomography, and Radiosonde in the dry period, 2015.

Figure 6 shows the statistics of the bias, STD, and RMS of the Tomography, the Output1, and the Output2 validated by the Radiosonde at different heights. In the wet period, bias of the Output1 is smaller than that of the Output2, but the differences are not obvious in terms of STD and RMS. In the dry period, the bias of the Output1 in the lower troposphere is slightly greater than that of the Output2. Overall, the differences between the Output1 and the Output2 are not significant.

In the wet period, the bias, STD, and RMS of the Tomography are greater than that of the Output1 in most of the time. But in the dry period, the STD and RMS of the Tomography tend to be smaller than that of the Output1 in the lower troposphere, but its bias is still greater. In general, the WRFDA performs better than the tomography

technique in most of the cases, but the RMS of Tomography validated by the Radiosonde in 400 m, 1600 m and
2400 m height is smaller than WRFDA output as shown in Figure 6f. So, in the lower troposphere in the dry
period the tomography performed better than the WRFDA in terms of RMS.

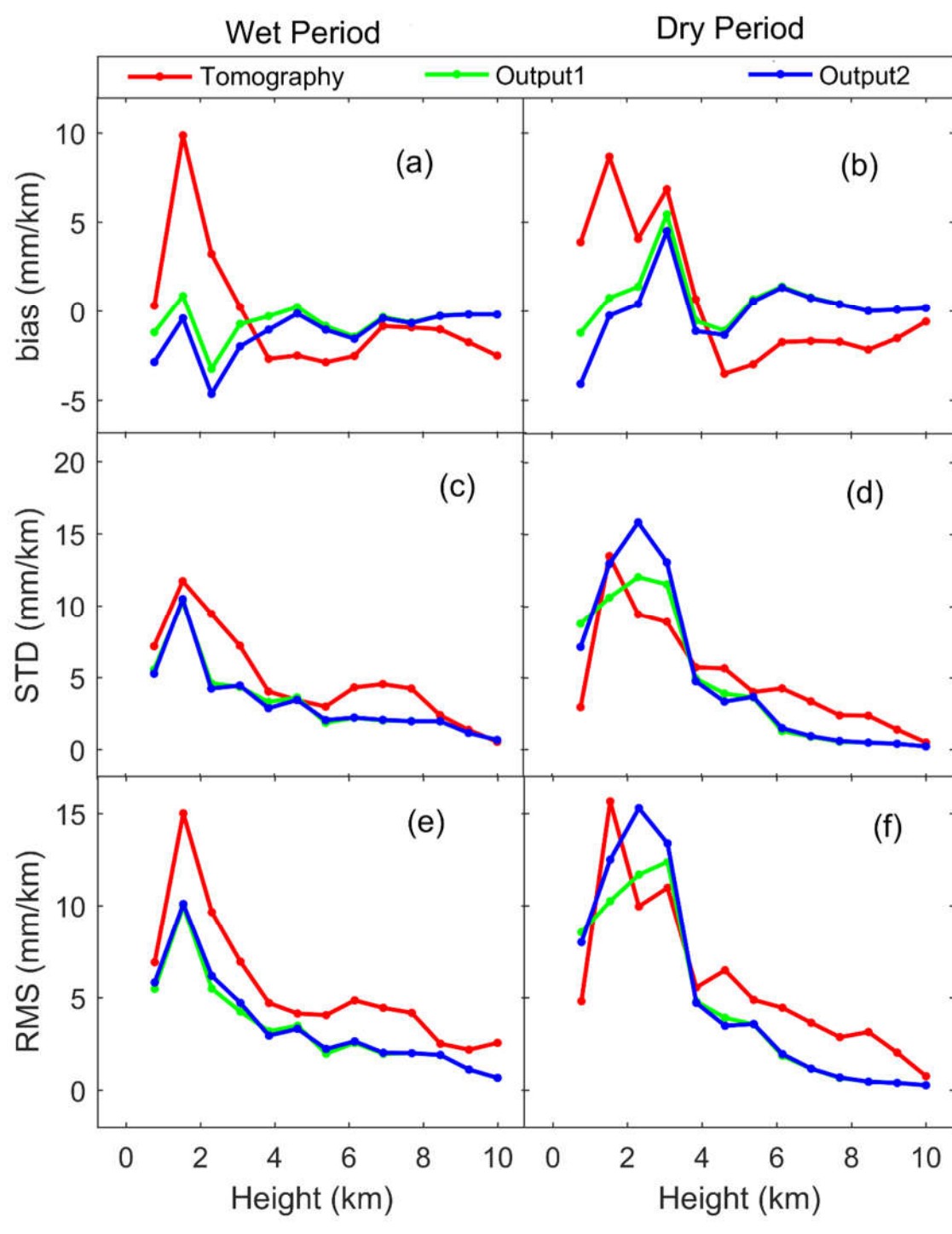


**Figure 6.** Statistics of bias, STD, and RMS of Tomography, Output1, and Output2 validated by the Radiosonde.
Table 2 shows the bias, STD, and RMS of the Tomography, the Output1, and the Output2 validated by the
Radiosonde. In the whole troposphere in the wet period, the Tomography has the smallest bias but the largest
STD and RMS. The Output1 and the Output2 have the similar STD and RMS that are much smaller than that
of the Tomography. But the Output2 has the largest bias than the Output1 and the Tomography. In the lower
troposphere in the wet period, the Output1 has the smallest STD and RMS while the Tomography has the largest
ones. The bias of Tomography is positive in the low troposphere but negative in the upper troposphere, this
should be due to the vertical smoothing constraints imposed on the WR. In the upper troposphere in the wet
period, Tomography has the largest bias, STD, and RMS while the Output1 has the smallest ones. Overall, both
the tomography and the WRFDA results have larger bias, STD, and RMS in the lower troposphere than in the
upper troposphere, indicating both the tomography technique and the data assimilation technique has deceased
capabilities in the lower troposphere.
In the whole troposphere in the dry period, the Output2 has the smallest bias but the largest STD and RMS. The
STD and RMS of the Tomography are larger than the Output1. In the lower troposphere in the dry period, the
Output2 has the largest RMS and STD while the Output1 as the smallest ones. In the low troposphere in the dry
period, the performance of the Tomography is not as good as the Output1 in terms of RMS. However, in the
upper troposphere in the dry period, the Tomography has relatively larger bias, STD and RMS than the WRFDA
results.
**Table 2.** Statistics of bias, RMS and STD of Tomography, Output1 and Output2 validated by the
radiosonde WR. Unit is mm/km.

| | | Wet Period | | | Dry Period | | |
|---|---|---|---|---|---|---|---|
| | | bias | STD | RMS | bias | STD | RMS |
| Total | Output1 | -0.64 | 4.11 | 4.15 | 0.63 | 6.34 | 6.35 |
| | Output2 | -1.19 | 4.15 | 4.31 | 0.10 | 7.28 | 7.26 |
| | Tomography | -0.31 | 6.51 | 6.50 | 0.63 | 7.01 | 7.02 |
| Low (< 5.6 km) | Output1 | -0.74 | 5.37 | 5.40 | 0.77 | 8.62 | 8.61 |
| | Output2 | -1.73 | 5.37 | 5.62 | -0.19 | 9.90 | 9.85 |
| | Tomography | 0.80 | 8.20 | 8.19 | 2.52 | 8.83 | 9.13 |
| Upper (≥ 5.6 km) | Output1 | -0.51 | 1.75 | 1.81 | 0.47 | 0.86 | 0.97 |
| | Output2 | -0.55 | 1.77 | 1.84 | 0.45 | 0.91 | 1.01 |
| | Tomography | -1.60 | 3.26 | 3.62 | -1.57 | 2.63 | 3.05 |

In general, assimilating GNSS ZTD into the WRFDA has slightly improved the WR retrieval by decreasing the
RMS by 0.2 mm/km. The WR derived from the Output1 and the Output2 has apparently smaller RMS than the
tomographic WR (4.15 mm/km vs. 6.50 mm/km and 4.31 mm/km vs. 6.50 mm/km, respectively). The results
obtained from WRFDA and tomography are better in the wet period than in the dry period, which is mainly due
to the sharp vertical variation of WR in the dry period.

**5 Discussion**

In the dry period, due to the sharp vertical variations of WR, the Tomography, the Output1 have decreased
performance in retrieving the WR, especially in the lower troposphere. Compared with the results in the wet
period, the RMS of the Tomography and the Output1 increases by 0.94 mm/km, 3.24 mm/km in the dry period,
respectively. The accuracy decrease is more significant in the Output1 than in the Tomography, resulting in that
the tomographic WR becomes better than the Output1 (Figures 6d and 6f) in the low troposphere.
When assimilating ZTD into the WRFDA, we only use the total water vapor and cannot use the vertical profile
of water vapor. This leads to that the assimilation of ZTD has limited improvement in retrieving the vertical
structure of the WR. Therefore, it is natural to consider assimilating the tomographic WR into the WRFDA to
improve the retrieval of the vertical structure of WR. At present, WRFDA could not assimilate WR directly,
but can assimilate meteorological parameters such as relative humidity, temperature and pressure. To assimilate
the tomographic WR, we convert WR to relative humidity.
The relationship between relative humidity (*RH*) and $P_w$ is shown as Equation (7).

$$RH = \frac{P_w}{P_s} \tag{7}$$

where $P_s$ is the saturated water vapor pressure which is related to temperature and can be calculated by Wexler
formula (Wexler, 1976,1977). The $P_w$ is calculated by Equation (1). The needed temperature and pressure
data are from the Output2.
After converting the tomographic WR to *RH*, we assimilate the *RH* together with the corresponding temperature
and pressure into the WRFDA. Then, the similar procedures as described in Section 3.1 are performed to
generate new WRFDA output.
The Tomography agrees better with the Radiosonde than the Output1 and the Output2 in the lower troposphere
below 3 km at 12:00 on August 6 (Figure 5l) and at 12:00 on August 7 (Figure 5n). So, we assimilate the
tomographic WR below 3 km into the WRFDA at these two epochs. The generated output data are denoted as
"Output3". The difference between Output3 and Radiosonde is denoted as "DA-Tomo". The difference between
Output1 and Radiosonde is denoted as "DA-ZTD". The difference between Tomography and Radiosonde is
denoted as "Tomo". The MAE at different heights at 12:00 on August 6 and 7 are shown in Figure 7.

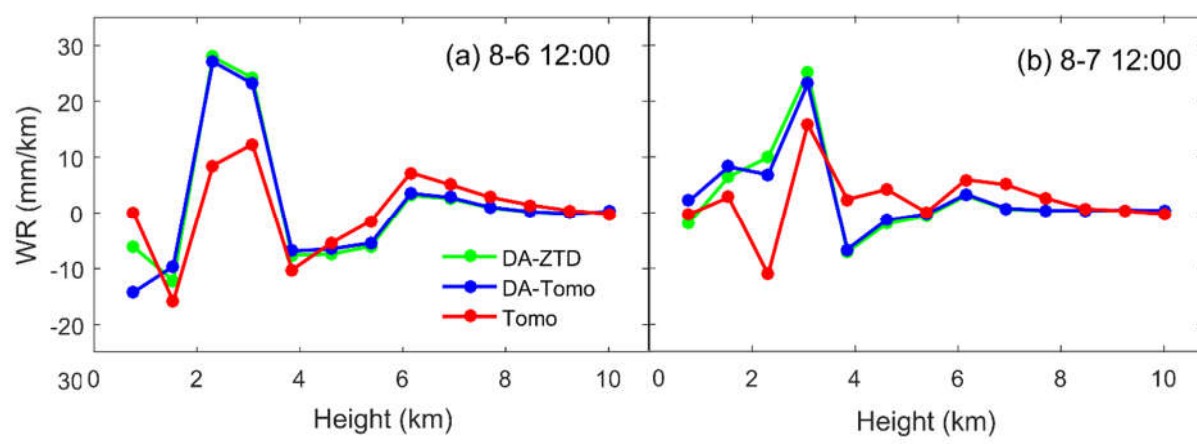


**Figure 7.** Differences between WR obtained by various methods and radiosonde WR.
Figure 7 shows that the DA-ZTD is very close to the DA-Tomo. The MAE of DA-ZTD is 6.04 mm/km and the
MAE of DA-Tomo is 5.92 mm/km. This indicates that assimilating tomographic WR into the WRFDA can
slightly improve the WR retrieve. But the large uncertainty (8.35 mm/km) of tomography WR in the lower
troposphere limit the improvement.
**6 Conclusions**
GNSS WR tomography and data assimilation experiments are conducted in Hong Kong during a wet and a dry
period to test the capabilities of the tomography technique and the WRFDA in retrieving WR. The results show
that both the tomography technique and the data assimilation technique can retrieve WR that agrees well with
the radiosonde data.
In the wet period in the whole troposphere, the RMS of Tomography, the Output1 and the Output2 are 6.50
mm/km, 4.31 mm/km, and 4.15 mm/km. The RMS becomes 7.02 mm/km, 6.35 mm/km, and 7.26 mm/km in
the dry period. Both methods obtained better WR in the wet period than in the dry period. We infer that the
sharp vertical variations of WR reduced the WR retrieving accuracy in the dry period. In most of the cases, the
Output1 outperforms the tomographic WR but the tomographic WR is better than the Output1 in the lower
troposphere in the dry period. By assimilating better tomographic WR in the lower troposphere into the WRFDA,
we slightly improve the retrieved WR.
The above results suggest that both the WRFDA and the tomography technique can retrieve good WR but also
have drawbacks. If we combine the two by assimilating good tomographic WR into the WRFDA, we may
further improve the performance of the WRFDA in retrieving the water vapor field.

*Data availability*. All the data used in this paper are available upon request by email (sggzb@whu.edu.cn).
**Appendix A**
The GNSS observation data are processed by the precise point positioning module in Bernese 5.0 software
using the same settings as detailed in Zhang et al. (2017). The International GNSS Service final orbit and clock
products are used. The differential code Biases (DCB) is corrected by products from the Center for Orbit
Determination in Europe. Antenna phase center offsets and variations, phase wind-up, Earth tides, Earth rotation,
ocean tides and relativistic effects are corrected by conventional methods detailed in (Kouba and Héroux, 2001).
We use the ionosphere-free combination of double frequencies to eliminate the first order ionospheric delay
and the higher-order terms are ignored. The tropospheric delay models are Saastamoinen model (Saastamoinen,
1972) and Niell mapping functions (Niell, 1996). The cut-off elevation angle is 10°. The station coordinates
and ZTDs are estimated simultaneously. Accurate zenith hydrostatic delays (ZHD) are estimated by using the
in-situ pressure observations and Saastamoinen model. The ZWD is estimated by removing the ZHDs from the
corresponding ZTDs. The SWD is reconstructed by mapping the ZWD and horizontal gradients onto the ray
direction. The phase residuals are added to SWD to consider the inhomogeneity of the troposphere.
*Conflicts of Interest*. The authors declare no conflict of interest.
*Acknowledgments*. The authors would like to thank the Survey and Mapping Office/Lands Department of Hong Kong,
IGRA, and ECMWF for providing experimental data, and thank Mesoscale and Microscale Meteorology Laboratory of
the National Center for Atmospheric Research for providing WRF model. This research is funded by the National Science
Foundation of China (41704004; 41574028), the Science Fund for Creative Research Groups of the National Natural
Science Foundation of China (41721003) and supported by Key Laboratory of Geospace Environment and Geodesy,
Ministry of Education, Wuhan University (17-02-03).

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
