# Peer review of "Comparisons between the WRF data assimilation and the GNSS tomography technique in retrieving 3D wet refractivity field in Hong Kong"

_Annales Geophysicae, 2018_

## Referee Comment (RC1) · Anonymous Referee #1 · 6 Sep 2018

General comments: The article is interesting in combining the two ways of retrieving 3D wet refractivity field. But it could still become significant with major changes, revisions. Therefore, the paper has the following issues that need to be addressed before it can go through the next process. The problems include (but are not limited to) the following.

Specific comments: - lines 46-51: I agree that 1) the least squares scheme or the Kalman filter scheme with additional constraints or using a priori information and 2) the algebraic reconstruction algorithm or similar methods are the main ways to get the results of the tomographic inversion, but the GNSS WV tomography methods quite a bit more than the tomographic inversion. The methods to establish the tomographic

region, and then to divide up this region by tomographic voxels or tomographic nodes are also very important for improving the tomographic model. For completeness, it would be of great benefit to the readers to say something about the other aspects of the tomographic method except the tomographic inversion.

- lines 85-96: I think that the process of the GNSS data does not need to be described in such detail. The parameter settings for Bernese 5.0 software can be presented by a table in the appendix. It may not be appropriate to devote much space in the main body of the paper.

- lines 103-104: Do I understand it correctly that the settings of the horizontal resolution and the number of the vertical layers are defined manually? If I understand it correctly then I am not clear the actual resolutions both in horizontal and vertical of the input data. I think it would be better to provide more information about the input data.

- line 122: Are you sure that the unit of specific humidity is kg/kg.

- section 3.2: Although a reference to GNSS tomography is useful, how to achieve this also an important content for readers in this paper. The detail information about the process of the GNSS tomography is needed to add.

- lines: 128-130: Priori information, as the initialization parameters, is the main part of the tomographic model. Line 144 shows that the background data (ECMWF ERA-Interim Data) has been used in WRF preprocessing system. Some background data as NWM fields or even standard atmosphere water vapor distribution can also be used in GNSS tomography. So I think it would be better to increase a set of experiments for the GNSS tomography with the priori information (GTPI). It would be interesting to compare the results from GTPI with other experiments which were conducted in the paper.

- lines: 139-140: You state that the reanalysis results and the radiosonde observation are interpolated to the centers of the associated tomography voxels. I am a little bit

surprised by this statement. The radiosonde (RS) data are used to validate the WR derived from GNSS tomography and reanalysis in this paper. Therefore, you should avoid adjusting the original data of RS. The other reason is that the RS data also exists the potential error. Comparison of the tomographic results with corresponding interpolation results of the RS may increase the influence of RS error due to insufficient results of the comparison. To get more reliable and complete results, it would be nice to interpolate the results of GNSS tomography and reanalysis to the position of the RS. On the one hand, can you explain how did you unify the units of the layers between the reanalysis and the GNSS tomography? You don't provide any information about it. I mean that the atmosphere is vertically divided into 45 layers by different pressure for the reanalysis (in line 104). However, for the GNSS tomography, the troposphere is vertically divided into 13 layers with a constant thickness of 800 meters (in line 131). It should be followed by one paragraph description of the method to unify the units.

- line 222: "time points" is not suitable for this case. The inversion of the tomographic WR needs GNSS data during a specific period of time. It's more like epochs than time points.

- lines 229-230: The average DA-ZTD and DA-Tomo may not suit for this case. It would be better to use Mean Absolute Error (MAE) instead of the average. MAE can avoid the canceling effect of the positive and negative values.

- lines 238-241: I am not sure that the rainy period and the rainless period present a striking contrast. Even in the rainy period, the WR derived from GNSS tomography and reanalysis only at the specific period (i.e., 0:00 and 12:00 UTC daily), at which it may not rain.

---

## Referee Comment (RC2) · Anonymous Referee #2 · 14 Sep 2018

General comments: The article shows an interesting study by combining two different approaches of retrieving 3-dimensional wet refractivity fields. However, in my opinion there are some major deficiencies, which need to be addressed before publication. What is the novelty of your approach and how can the NWP community benefit from this? This is not fully clear especially with respect to the huge effort in creating the tomography fields compared to a simple ZTD calculation. Therefore, I suggest major revisions before publication of the manuscript. Please avoid using "may" all the time. Please make a more precise statement.

As I am not an expert in GPS tomography, I will mainly focus on the modeling part.

Specific comments:

lines 55-62: Are there also other NWP models than WRF, which make use of ZTD/PWV data assimilation?

line 77: What do you mean with "vertically flat" in this case. Is your statement related to the altitude difference of 344 m? I do not think that this can be considered as "flat".

Line 84: maybe "dry" instead of "rainless".

Lines 85-96: I think it is a good idea to show the applied parameters for Bernese in a separate table. In line 92/93, I guess you mean Niell in both cases. Are all GNSS receivers equipped with temperature and pressure measurements? If not, please mention how you derive the ZTD at the receiver locations.

Lines 99-102: The general purpose of any data assimilation scheme is to obtain the best estimate of the atmosphere not only with respect to ZTD observations. Did you assimilate any other observations than ZTD? It is well known, that one should make use of all available observations to complement each other. Especially as the 3DVAR does not contain any dynamical component. How was the 3DVAR set up? Is it a rapid update cycle with e.g. an hourly update or did you ran the 3DVAR once at the beginning of your period of interest? Did you apply multiple outer loops? What is the ZTD error you used? All these details are important to know as this determines the weight/impact of the observations and the data assimilation in general.

Lines 102-104: Does the model domain only encompass the area shown in Figure 1? If this is the case, you may only have approx. 30*25 grid cells. Assuming a boundary relaxation zone of 5 cells, you effective model domain will be 20*15 cells which is far too small. The model does not have a chance to develop its own state but is mainly determined by the boundary conditions. Please clarify. Did you apply the default layer settings in WRF by setting "eta_levs" to a certin value or did you define the levels on your own? How many layers are in the PBL? This may be important as the majority of

the humidity is located inside the PBL. A lot more information is necessary here.

Lines 109-111: Did you apply the reanalysis of ERA-Interim, which has a resolution of $0.75°$ and not $0.125°$ or the operational analysis? Are the forcing data applied on model levels or on pressure levels? In case you applied the former data, I'm afraid that this is not a suitable data to study the behaviour of a convection permitting model especially at these short time scales although data assimilation is applied.

Line 115: To me the applied CV3 method is a major concern. I guess you know that this matrix is derived from a NCEP model climatology at a horizontal resolution of roughly $2°$ and this is applied on the CP scale in your study. I am concerned if this is a scientifically valid approach.

Line 118: I think the word reanalysis is misleading here as you probably only used ZTD observations. I also do not really see from the Vedel and Huang publication how WR is derived. Please also include the units for k1 and k2. Are T and P only used at the surface? This is not clear here. Also, please use "p" instead of "P" for pressure.

Line 137: Do you mean the ECMWF (re-)analysis or the new analysis obtained from WRFDA?

Line 138: What do you mean with "nearest four grids"?

Lines 139-140: Why did you adjust the radiosonde data? This distorts the radiosonde observations. I strongly recommend to interpolate the GNSS and tomography fields to the radiosonde location. How did you interpolate the unevenly distributed WRF model layers to the tomography layers, which have a constant spacing? The native WRF model output is not on pressure levels but on terrain following coordinates. I think is it necessary to include a short paragraph here.

Line 143: Is "Reanalysis 2" your control run mentioned in line 117 or is this an assimilation run where everything except ZTDs was assimilated?

Line 152: Why does this lead to a decrease in performance in the tomography and the
WRF model?

Line 154: Did you perform any significance tests or do you mean something like "considerably"?

Line 173: Why does the tomography "may" perform better than WRF? I though you did investigate this?

Lines 196-198: This statement is very confusing and queries the results of your study.

Line 208: With ZTD you do not assimilate the column water vapour. The signal delay is assimilated from which TCW can be derived.

Line 210: From your results I do not agree with this statement.

Line 213: How did you assimilate relative humidity only in WRFDA? If you use p and T from reanalysis (again the WRFDA analysis or ECMWF?), the tomography is not model independent anymore.

---

## Author Comment (AC1) · 18 Oct 2018

Response to Reviewer #1:

Q1: lines 46-51: I agree that 1) the least squares scheme or the Kalman filter scheme with additional constraints or using a priori information and 2) the algebraic reconstruction algorithm or similar methods are the main ways to get the results of the tomographic inversion, but the GNSS WV tomography methods quite a bit more than the tomographic inversion. The methods to establish the tomographic region, and then to divide up this region by tomographic voxels or tomographic nodes are also very important for improving the tomographic model. For completeness, it would be of great

benefit to the readers to say something about the other aspects of the tomographic method except the tomographic inversion.

Response: Thanks for the suggestion. We added the literature review about other aspects of the tomographic method, including voxel division, the usage of virtual reference stations, and using the rays coming from the side face of the tomography area. The details are as follows (copied from lines 54-57): "Besides the algorithm improvement, some scientists tried to optimize the voxel division (Chen and Liu, 2014) or use virtual reference stations (Adavi and Mashhadi-Hossainali, 2014) or use additional GNSS rays (Zhao and Yao, 2017) to increase the effective GNSS rays and thus improve the tomography results."

Q2: lines 85-96: I think that the process of the GNSS data does not need to be described in such detail. The parameter settings for Bernese 5.0 software can be presented by a table in the appendix. It may not be appropriate to devote much space in the main body of the paper.

Response: According to Reviewer's suggestion, we moved the description about the process of the GNSS data and the reconstruction of SWD to Appendix A. See lines 93 and 290-302.

Q3: lines 103-104: Do I understand it correctly that the settings of the horizontal resolution and the number of the vertical layers are defined manually? If I understand it correctly then I am not clear the actual resolutions both in horizontal and vertical of the input data. I think it would be better to provide more information about the input data.

Response: Yes, the settings of the horizontal resolution and the number of the vertical layers should be defined manually before running the WRF model. The main input data is the ECMWF ERA-interim reanalysis whose nominal resolution is 0.125° but actual resolution is 0.75°. This has been clarified in lines 112-113.

Q4: line 122: Are you sure that the unit of specific humidity is kg/kg.

Response: It is g/g, we have corrected it in line 126.

Q5: section 3.2: Although a reference to GNSS tomography is useful, how to achieve this also an important content for readers in this paper. The detail information about the process of the GNSS tomography is needed to add.

Response: As suggested, we have added the detailed description about our GNSS tomography algorithm in lines 130-166.

Q6: lines: 128-130: Priori information, as the initialization parameters, is the main part of the tomographic model. Line 144 shows that the background data (ECMWF ERA Interim Data) has been used in WRF preprocessing system. Some background data as NWM fields or even standard atmosphere water vapor distribution can also be used in GNSS tomography. So I think it would be better to increase a set of experiments for the GNSS tomography with the priori information (GTPI). It would be interesting to compare the results from GTPI with other experiments which were conducted in the paper.

Response: The tomography model does not necessarily depend on a priori information. The core idea of our tomography algorithm is free of a priori information (not count the constraints) and we try to make tomography algorithm an independent technique to retrieve 3D wet refractivity field. If a priori information from NWM is used, the solution will be highly correlated with the NWM data. The WRF output is largely determined by the background data (also NWM data). Consequently, the tomography solution and WRF output will be correlated. If we make such a comparison (tomography with NWM vs. WRF output), it will be unfair and less meaningful. We have clarified this in lines 132-134. In addition, the NWM data has a very coarse resolution (even the latest ERA-5 data have a horizontal resolution of ~30 km, basically one grid node covers our research area), which means it cannot reflect the water vapor distribution within our research area. Base on the above reasons, we choose not to do such a comparison.

Q7: lines: 139-140: You state that the reanalysis results and the radiosonde observation are interpolated to the centers of the associated tomography voxels. I am a little bit surprised by this statement. The radiosonde (RS) data are used to validate the WR derived from GNSS tomography and reanalysis in this paper. Therefore, you should avoid adjusting the original data of RS. The other reason is that the RS data also exists the potential error. Comparison of the tomographic results with corresponding inter-polation results of the RS may increase the influence of RS error due to insufficient results of the comparison. To get more reliable and complete results, it would be nice to interpolate the results of GNSS tomography and reanalysis to the position of the RS. On the one hand, can you explain how did you unify the units of the layers between the reanalysis and the GNSS tomography? You don't provide any information about it. I mean that the atmosphere is vertically divided into 45 layers by different pressure for the reanalysis (in line 104). However, for the GNSS tomography, the troposphere is vertically divided into 13 layers with a constant thickness of 800 meters (in line 131). It should be followed by one paragraph description of the method to unify the units.

Response: Thank you for your suggestion. In the vertical troposphere, the tomography model has only 13 layers whose vertical resolution is only 800 m while the radiosonde has a vertical resolution of $\sim$23 layers from 0 km to 10 km height. It means that the radiosonde data have a much better vertical resolution than the tomography results. Therefore, we think interpolating the dense radiosonde data to the sparse tomogra-phy layers in the vertical direction would be more accurate. We show the original radiosonde profiles in Figures 3 and 4 now. We convert the pressure levels to geopo-tential height by NCL. The heights of the tomography layers are converted to normal height. We neglect the slight difference between the geopotential height and normal height. By these efforts, we unify the units of the layers and make the different results comparable. This has been clarified in lines 170-175.

Q8: line 222: "time points" is not suitable for this case. The inversion of the tomographic WR needs GNSS data during a specific period of time. It's more like epochs than time points.

Response: Thank you for your suggestion. And we replaced the "time points" with "epochs" in the manuscript.

Q9: lines 229-230: The average DA-ZTD and DA-Tomo may not suit for this case. It would be better to use Mean Absolute Error (MAE) instead of the average. MAE can avoid the canceling effect of the positive and negative values.

Response: Thank you for your suggestion. Actually, the DA-ZTD and DA-Tomo are MAE in the manuscript. This has been clarified in lines 264, 268-269 in the manuscript.

Q10: lines 238-241: I am not sure that the rainy period and the rainless period present a striking contrast. Even in the rainy period, the WR derived from GNSS tomography and reanalysis only at the specific period (i.e., 0:00 and 12:00 UTC daily), at which it may not rain.

Response: Thank you for your suggestion. We replaced the 'rainy' with 'wet' in the manuscript. During July 20 to 26, it did not keep rainning all the time, but it was really wet. And the wet and dry (rainless) period indeed show different vertical distribution of the water vapor.

Please also note the supplement to this comment:
https://www.ann-geophys-discuss.net/angeo-2018-84/angeo-2018-84-AC1-supplement.pdf

**Supplement:**

[revised manuscript text omitted]

---

## Author Comment (AC2) · 18 Oct 2018

Response to Reviewer #2:

Q1: The article shows an interesting study by combining two different approaches of retrieving 3-dimensional wet refractivity fields. However, in my opinion there are some major deficiencies, which need to be addressed before publication. What is the novelty of your approach and how can the NWP community benefit from this? This is not fully clear especially with respect to the huge effort in creating the tomography fields compared to a simple ZTD calculation.

[Figure]

Response: The novelty of this manuscript is (1) we use an advanced tomography approach to retrieve the 3D wet refractivity filed; (2) we conduct a fair comparison between the tomography technique and the WRF model, which is seldomly done by the NWP community or the GNSS community. The benefits of this study are (1) provides insights for the NWP community about this new technique and the possibility of assimilating the tomography results into the NWP models; and (2) the GNSS community will get a better understanding of the WRF model and its capability in simulating the water vapor field. This has been clarified in lines 73-77.

Q2: lines 55-62: Are there also other NWP models than WRF, which make use of ZTD/PWV data assimilation?

Response: Yes, the AROME NWP system and Japan Meteorological Agency (JMA) Mesoscale Numerical Weather Prediction Model can also make use of ZTD\PWV data assimilation. We added the citations of the related models in lines 69-71. Here are the references: Nakamura H, Koizumi K, Mannoji N. Data assimilation of GPS precipitable water vapor into the JMA mesoscale numerical weather prediction model and its impact on rainfall forecasts[J]. Journal of the Meteorological Society of Japan. Ser. II, 2004, 82(1B): 441-452. Boniface K, Ducrocq V, Jaubert G, et al. Impact of high-resolution data assimilation of GPS zenith delay on Mediterranean heavy rainfall forecasting[C]//Annales Geophysicae. 2009, 27: 2739-2753.

Q3: line 77: What do you mean with "vertically flat" in this case. Is your statement related to the altitude difference of 344 m? I do not think that this can be considered as "flat".

Response: In GNSS tomography, a network whose altitude differences are less than 1 km is regarded as a flat network. Flat networks bring difficulties in retrieving the vertical solutions of the WR. We have clarified this in line 88.

Q4: Line 84: maybe "dry" instead of "rainless".

[Figure]

Response: Thank you for your suggestion. And we have used 'dry' instead of 'rainless' in the manuscript.

Q5: Lines 85-96: I think it is a good idea to show the applied parameters for Bernese in a separate table. In line 92/93, I guess you mean Niell in both cases. Are all GNSS receivers equipped with temperature and pressure measurements? If not, please mention how you derive the ZTD at the receiver locations.

Response: According to you and the other reviewer's suggestions, we have moved the description about the GNSS data processing to Appendix A (lines 289-302). Sorry, it's Niell, we made a typo and has corrected it. Yes, all the GNSS receivers are equipped with temperature, relative humidity, pressure measurements.

Q6: Lines 99-102: The general purpose of any data assimilation scheme is to obtain the best estimate of the atmosphere not only with respect to ZTD observations. Did you assimilate any other observations than ZTD? It is well known, that one should make use of all available observations to complement each other. Especially as the 3DVAR does not contain any dynamical component. How was the 3DVAR set up? Is it a rapid update cycle with e.g. an hourly update or did you ran the 3DVAR once at the beginning of your period of interest? Did you apply multiple outer loops? What is the ZTD error you used? All these details are important to know as this determines the weight/impact of the observations and the data assimilation in general.

Response: We delete the inaccurate expression "In this study, the WRFDA estimates the atmosphere state that best fits the ZTD observations." The purpose of this manuscript is to conduct an interesting and fair comparison between the tomography technique and the WRF model. To be fair, we use only GPS data for both tomography and the WRF model, i.e. slant wet delay for tomography and ZTD for the WRF model. In addition, except for the GPS data, only the surface meteorological observations can be assimilated into the WRF model (the only radiosonde data will be left for validation), but assimilating the surface meteorological data into WRF can make very little difference, according to our previous tests. The physics options are unified Noah land-surface model (Tewari et al., 2004), Revised MM5 Monin-Obukhov scheme (Monin and Obukhov, 1954), and Yonsei University planetary boundary layer scheme (Hong et al., 2006). The Rapid Radiative Transfer Model (Mlawer et al., 1997) and Dudhia's scheme (Dudhia, 1989) were used for longwave radiation and shortwave radiation, respectively. The physics settings of WRFDA are the same with WRF (This has been clarified in lines 104-110). This experiment does not apply multiple outer loops and just run the 3DVAR once at the beginning of the period of interest. The ZTD error is output by the Bernese 5.0 software.

Q7: Lines 102-104: Does the model domain only encompass the area shown in Figure 1? If this is the case, you may only have approx. 30*25 grid cells. Assuming a boundary relaxation zone of 5 cells, you effective model domain will be 20*15 cells which is far too small. The model does not have a chance to develop its own state but is mainly determined by the boundary conditions. Please clarify. Did you apply the default layer settings in WRF by setting "eta_levs" to a certin value or did you define the levels on your own? How many layers are in the PBL? This may be important as the majority of the humidity is located inside the PBL. A lot more information is necessary here.

Response: Yes, the model domain only encompasses the area shown in Figure 1. And the relaxing zone is 4 cells. We just need the model reanalysis at the beginning of the interested period, we don't need the model to develop its state in time. Namely, we just need the model uses the observations to update the background data. We set 46 layers in WRF on our own and 10 layers in the PBL.

Q8: Lines 109-111: Did you apply the reanalysis of ERA-Interim, which has a resolution of 0.75âŮę and not 0.125âŮę or the operational analysis? Are the forcing data applied on model levels or on pressure levels? In case you applied the former data, I'm afraid that this is not a suitable data to study the behaviour of a convection permitting model especially at these short time scales although data assimilation is applied.

Response: Yes, I applied the reanalysis of ERA-Interim. We use the ERA-Interim data on pressure levels and surface data. Its nominal resolution is $0.125° \times 0.125°$ and the real resolution is $0.75° \times 0.75°$.

Q9: Line 115: To me the applied CV3 method is a major concern. I guess you know that this matrix is derived from a NCEP model climatology at a horizontal resolution of roughly 2âŮę and this is applied on the CP scale in your study. I am concerned if this is a scientifically valid approach.

Response: The ARW version 3 Modeling System User's Guide (published in July 2016) says that "However, CV3 (a BE file provided with our WRFDA system) is a global BE and can be used for any regional domain, while CV5, CV6, and CV7 BE's are domain-dependent, and so should be generated based on forecast or ensemble data from the same domain." in page 6-39. "Theoretically, CV3 BE is a generic background error statistics file which can be used for any case." in page 6-40. Based on these, we adopted the default CV3 background error. It may not be the best, but it could be used in this case.

Q10: Line 118: I think the word reanalysis is misleading here as you probably only used ZTD observations. I also do not really see from the Vedel and Huang publication how WR is derived. Please also include the units for k1 and k2. Are T and P only used at the surface? This is not clear here. Also, please use "p" instead of "P" for pressure.

Response: Thank you for your suggestion. And we revise the term 'reanalysis' to 'output'. Vedel and Huang (2004) didn't directly give the equation for WR calculation, but we can easily partition the equation for WR calculation from the equation for wet delay calculation. We attach a figure (Fig.1, in which the code is from da_transform_xtoztd.inc in WRFDA) to show Vedel and Huang's equation. In the equation, wzd is the zenith wet delay, which is calculated by integrating the product of WR and altitude difference (which represented by dh). Therefore, the wet refractivity can be easily derived by dividing the wzd increment by the dh. The unit of k1 is K/Pa, and the unit of k2 is $K^2$/Pa. We clarified this in the manuscript in line 124. We have replaced "P" with "p".

Q11: Line 137: Do you mean the ECMWF (re-)analysis or the new analysis obtained from WRFDA?

Response: The reanalysis means WRFDA and WRF output, namely the new reanalysis.

Q12: Line 138: What do you mean with "nearest four grids"?

Response: This part has been rewritten, no "nearest four grids" any longer. The new expressions are: "The vertical coordinates of WRF model output are converted to geopotential heights by NCL and the geodetic heights of tomographic results are converted to normal height. The slight differences between geopotential heights and normal heights are neglected. We interpolate the WRF output to tomographic nodes since the former has a much higher resolution (23 layers from 0 to 10 km height) than the latter (13 layers) and thus we can get a higher interpolation accuracy. We use a bilinear interpolation method in the horizontal domain and a linear interpolation method in the vertical direction.".

Q13: Lines 139-140: Why did you adjust the radiosonde data? This distorts the radiosonde observations. I strongly recommend to interpolate the GNSS and tomography fields to the radiosonde location. How did you interpolate the unevenly distributed WRF model layers to the tomography layers, which have a constant spacing? The native WRF model output is not on pressure levels but on terrain following coordinates. I think is it necessary to include a short paragraph here.

Response: Thank you for your suggestion. In the vertical troposphere, the tomography model only has 13 layers whose vertical resolution is only 800 m while the radiosonde has a vertical resolution of ∼23 layers from 0 km to 10 km height. It means the radiosonde data have a much better vertical resolution than the tomography re-

sults. Therefore, we think interpolating the dense radiosonde data to the sparse to-mography layers in the vertical direction would be more accurate. We show the original radiosonde profiles in Figures 3 and 4 now. The vertical coordinates of WRF model output are converted to geopotential heights by NCL and the geodetic heights of tomographic results are converted to normal height. The slight differences between geopotential heights and normal heights are neglected. We interpolate the WRF output to tomographic nodes since the former has a much higher resolution than the latter and thus we can get a higher interpolation accuracy. We use a bi-linear interpolation method in the horizontal domain and linear interpolation method in the vertical direction. By these methods, we interpolate both WRF output and radiosonde results to the tomography nodes. This has been illustrated in lines 170-177 in the manuscripts.

Q14: Line 143: Is "Reanalysis 2" your control run mentioned in line 117 or is this an assimilation run where everything except ZTDs was assimilated?

Response: The "Reanalysis2" is the control run and assimilates nothing. This has been clarified in line 119-120.

Q15: Line 152: Why does this lead to a decrease in performance in the tomography and the WRF model?

Response: Figures 3 and 4 show that the WR was distributed evenly from 0 to 10 km in July period when Hong Kong rains heavily, while the WR concentrated from 0 to 6 km in August period. This means that the water vapor varied sharply in the vertical direction in the August period and was relatively smooth in the July period. Both the model and the tomography technique can better retrieve the water vapor with smooth distribution than that with sharp variations. This has been clarified in lines 185-189.

Q16: Line 154: Did you perform any significance tests or do you mean something like "considerably"?

Response: No, we did not. We treat the radiosonde data as the true values and use them to validate the other results. The expression "Compared with Reanalysis2, the Reanalysis1 is slightly improved, but the improvement is not significant" is kind of misleading, we have revised it to "Compared with Reanalysis2, the Reanalysis1 is slightly improved by reducing the MAE by 1.25 mm/km". This has been revised in lines 190-192.

Q17: Line 173: Why does the tomography "may" perform better than WRF? I though you did investigate this?

Response: The RMS of the tomography results is smaller than the Reanalysis1 results at 400 m, 1600 m, and 2400 m height, which is shown in Figure 5f. From the statistical perspective, the tomography performs better than the WRF model at these heights. We have clarified this and deleted the term "may", see lines 209-210.

Q18: Lines 196-198: This statement is very confusing and queries the results of your study.

Response: To be more specific and accurate, we have revised the whole paragraph to (lines 234-238): "In general, assimilating GNSS ZTD into the WRF model has slightly improved the WR retrieval by decreasing the RMS by 0.2 mm/km. The WR derived from Reanalysis1 and Reanalysis2 has apparently smaller RMS than the tomographic WR (4.15 mm/km vs. 6.50 mm/km and 4.31 mm/km vs. 6.50 mm/km, respectively). The results obtained from WRF and tomography are better in the wet period than in the dry period, which is mainly due to the sharp vertical variation of WR in the dry period."

Q19: Line 208: With ZTD you do not assimilate the column water vapour. The signal delay is assimilated from which TCW can be derived.

Response: Yes, we just need to assimilate one of them.

Q20: Line 210: From your results I do not agree with this statement.

Response: Make full use of the vertical structure information of water vapor could benefit the data assimilation. It could provide more information such as the vertical water vapor distribution than ZTD. And the MAE of assimilating ZTD is 6.04 mm/km while the MAE of assimilating tomographic WR is 5.92 mm/km. This result shows that assimilating tomographic WR could improve the WR retrieve.

Q21: Line 213: How did you assimilate relative humidity only in WRFDA? If you use p and T from reanalysis (again the WRFDA analysis or ECMWF?), the tomography is not model independent anymore.

Response: According to the WRF data assimilation scheme, we can only assimilate the relative humidity together with T and p from WRF output. We tried to mitigate model dependence. We use the T and P from the WRF output (O1) without assimilating anything. Then we run WRFDA to assimilate the tomographic WR together with T and p, and then run WRF to generate the new output (O2). The T and P in O1 and O2 have very small difference, therefore we controlled the influence of T and P on the final results. The difference between O1 and O2 mainly lies in the humidity, which was caused only by the tomographic WR.

Please also note the supplement to this comment:
https://www.ann-geophys-discuss.net/angeo-2018-84/angeo-2018-84-AC2-supplement.pdf
* * *
```
subroutine da_transform_xtoztd(grid)

!------------------------------------------------------------
!  Purpose: to compute the Zenith Total Delay, and save it to xb%ztd.
!
!  Both of the wet and dry delay are computed based on Vedel and Huang,
!         J. Meteor. Soc., 82, 459-472, 2004.
!
!      ** Equation (3) in Vedel and Huang is wrong.
!
!             ported by Yong-Run Guo  05/12/2008 from wrf3dvar.
!------------------------------------------------------------

   implicit none

   type (domain), intent(inout) :: grid

   integer :: i, j, k

   real    :: const, part, term1, term2, wzd, hzd, zf

   if (trace_use) call da_trace_entry("da_transform_xtoztd")

!--WEIGHTED SUM OF VERTICAL COLUMN
   do j=jts, jte
      do i=its, ite

! Wet delay:
      wzd = 0.0
      do k=kts, kte
         const = (grid%xb%hf(i,j,k+1)-grid%xb%hf(i,j,k)) / a_ew
         part  = grid%xb%p(i,j,k)*grid%xb%q(i,j,k) / grid%xb%t(i,j,k)
         term1 = part * const * wdk1
         term2 = part * const * wdk2 / grid%xb%t(i,j,k)
         wzd   = wzd + term1 + term2
      enddo

! Hydrostatic delay (Saastamoinen 1972):
      zf = (1.0 - zdk2*cos(2.0*grid%xb%lat(i,j)*radian) - zdk3*grid%xb%terr(i,j))
      hzd = zdk1 * grid%xb%psfc(i,j) / zf
```

**wzd = Σ WR * dh**

**dh**

**Fig. 1.** Fig1

**Supplement:**

[revised manuscript text omitted]

---

## Author Comment (AC3) · 30 Nov 2018

Dear Editor,

Thank you very much for your comments and suggestions. We have carefully revised the manuscript accordingly. The revisions and responses are listed below.

Q1: Regarding reviewer #2, Question 6: I think what might be meant here is that one could for example assimilate wind. Water vapor could then still be used for evaluation.
**Response:** The purpose of this manuscript is to conduct an interesting and fair comparison between the WRFDA model and the tomography method in retrieving wet refractivity. For the sake of fairness, both techniques use only the GPS troposphere delay products. At current stage, we only want to know how the much improvement can be achieved by assimilating the GPS tropospheric delay products into the WRFDA model.

Q2: I also do not understand why 3DVAR was applied only once at the beginning of the period of interest.
**Response:** Because we just need the data assimilating results at certain epochs (0:00 and 12:00 UTC) and these epochs are also the beginning of the interested periods. We have the validating data (radiosonde data) only at 0:00 and 12:00 UTC, therefore we just need the assimilating results and tomography results at these epochs. We start running the WRFDA model at 0:00 and 12:00 UTC, so we only need to run 3DVAR once.

Q3: Regarding reviewer #2, Question 7: since not only WRFDA is used, but also WRF is run, I do not understand why you "don't need the model to develop its state in time". If you are using WRF output and the domain is too small to represent the relevant scales, in my opinion, this could very well be a major problem.
**Response:** We find that we caused a lot of confusion by running the WRF model. Previously, we first run WRFDA 3DVAR to assimilate ZTDs and generate the output, labeled as output1. Then, we run the WRF using output1 as the background value to generate output2 at the same epoch. Actually, we just want to do data assimilating other than hindcasting, we don't need to run WRF to generate output2. I think running the WRF after 3DVAR caused a lot of confusion, we have removed the WRF part from the manuscript and redone the validation using output1 instead of output2. By these revisions, we have focused on the data assimilation and the WRFDA model.

Q4: In principle, I think that it would be good to find out how sensitive the results are to the domain layout (size, position, and resolution). Furthermore, WRF provides a large choice of physics parameterizations, and choosing specific parameterizations could also impact the results.
**Response:** Now, we only run WRFDA in the manuscript. The WRFDA has many options for different physical parameterizations. In order to find the best choice for the data assimilation experiment, we follow Chien et al. (2006) to set the schemes as listed in Table 1. We carry out the sensitivity test at 00:00 UTC $22^{nd}$ July in 2015. The initial

domain size is set to 30 × 24 grids which just cover the study area. The grid size is 3 km × 3km. Then, we run WRFDA using the different setting schemes. The radiosonde data are used to validate the wet refractivity derived by the WRFDA output. Table 1 shows that all schemes for WRFDA has the same bias, standard deviation (STD), and Root Mean Square (RMS). It appears that the output wet refractivity is not affected by the physical parameterization settings in WRFDA.

**Table R1.** Physical parameterization schemes and statistics of bias, RMS and STD of wet refractivity validated by radiosonde data. Unit is mm/km.

|  | PBL physics | cumulus physics | microphysics | bias | STD | RMS |
|---|---|---|---|---|---|---|
| 1 | Yonsei University | Kain-Fritsch | WSM 5-class | -3.95 | 6.55 | 7.51 |
| 2 | Yonsei University | Betts-Miller-Janjic | WSM 5-class | -3.95 | 6.55 | 7.51 |
| 3 | Yonsei University | Grell-Freitas ensemble | WSM 5-class | -3.95 | 6.55 | 7.51 |
| 4 | Yonsei University | Kain-Fritsch | Ferrier | -3.95 | 6.55 | 7.51 |
| 5 | Yonsei University | Betts-Miller-Janjic | Ferrier | -3.95 | 6.55 | 7.51 |
| 6 | Yonsei University | Grell-Freitas ensemble | Ferrier | -3.95 | 6.55 | 7.51 |
| 7 | Mellor-Yamada-Janjic | Kain-Fritsch | WSM 5-class | -3.95 | 6.55 | 7.51 |
| 8 | Mellor-Yamada-Janjic | Betts-Miller-Janjic | WSM 5-class | -3.95 | 6.55 | 7.51 |
| 9 | Mellor-Yamada-Janjic | Grell-Freitas ensemble | WSM 5-class | -3.95 | 6.55 | 7.51 |
| 10 | Mellor-Yamada-Janjic | Kain-Fritsch | Ferrier | -3.95 | 6.55 | 7.51 |
| 11 | Mellor-Yamada-Janjic | Betts-Miller-Janjic | Ferrier | -3.95 | 6.55 | 7.51 |
| 12 | Mellor-Yamada-Janjic | Grell-Freitas ensemble | Ferrier | -3.95 | 6.55 | 7.51 |

Reference:

Chien F C, Hong J S, Chang W J, et al. A sensitivity study of the WRF model. Part II: verification of quantitative precipitation forecasts[J]. Atmos. Sci, 2006, 34(3): 261-276.

In order to figure out how sensitive the wet refractivity output is to the domain size, we carry out a sensitivity test at 00:00 UTC 22nd July in 2015. And we increase the domain size gradually from 30 × 24 grids to 190 × 184 grids. In each run, we validate the wet refractivity derived by the WRFDA output using the radiosonde data. The statistical results of the sensitivity test are shown in Figure 1. It shows that the smaller domain size has the smaller bias, STD, and RMS. So, the domain size of the data assimilation experiment is set to 30 × 24 grids which just cover the study area. This has been discussed in lines 115-138 in the manuscript.

[Figure]

**Figure R1.** Statistics of sensitivity test with different domain size.

Q5: l. 3, l. 34, l. 58, and elsewhere : filed -> field (and also fileds -> fields)
**Response:** Thank you. We revised it in the manuscript. Lines: 3, 31, 55, 66, 70, 71, 77, 79, 293.

Q6: l. 100: And the start time of the WRF model is epoch of interest. -> please be more specific
**Response:** We have revised it to "And we run the WRFDA model at 0:00 UTC and 12:00 UTC, corresponding to the radiosonde observation time." See lines 102-103.

Q7: l. 171: NCL: NCAR Command Language, I suggest to add a reference (see e.g. https://www.ncl.ucar.edu/citation.shtml)
**Response:** We added it in the manuscript. Lines: 183 and 410-411.

Q8: l. 180: I think WRF output1 and WRF output2 would be more accurate than reanalysis1, reanalysis2.
**Response:** Thank you, and we revised Reanalysis1 to Output1 and Reanalysis2 to Output2 in lines 109-110 in the manuscript.

---

## Author Comment (AC5) · 30 Nov 2018

Supplement the revised manuscript

Please also note the supplement to this comment:
https://www.ann-geophys-discuss.net/angeo-2018-84/angeo-2018-84-AC5-supplement.pdf

---

## Author Comment (AC6) · 30 Nov 2018

**Response to Reviewer #2:**

Q1: The article shows an interesting study by combining two different approaches of retrieving 3-dimensional wet refractivity fields. However, in my opinion there are some major deficiencies, which need to be addressed before publication. What is the novelty of your approach and how can the NWP community benefit from this? This is not fully clear especially with respect to the huge effort in creating the tomography fields compared to a simple ZTD calculation.

**Response:** The novelty of this manuscript is (1) we use an advanced tomography approach to retrieve the 3D wet refractivity filed; (2) we conduct a fair comparison between the tomography technique and the WRF data assimilation, which is seldomly done by the NWP community or the GNSS community. The benefits of this study are (1) provides insights for the NWP community about this new technique and the possibility of assimilating the tomography results into the NWP models; and (2) the GNSS community will get a better understanding of the WRFDA and its capability in simulating the water vapor field. This has been clarified in lines 71-75.

Q2: lines 55-62: Are there also other NWP models than WRF, which make use of ZTD/PWV data assimilation?

**Response:** Yes, the AROME NWP system and Japan Meteorological Agency (JMA) Mesoscale Numerical Weather Prediction Model can also make use of ZTD\PWV data assimilation. We added the citations of the related models in lines 67-69.
Here are the references:
Nakamura H, Koizumi K, Mannoji N. Data assimilation of GPS precipitable water vapor into the JMA mesoscale numerical weather prediction model and its impact on rainfall forecasts[J]. Journal of the Meteorological Society of Japan. Ser. II, 2004, 82(1B): 441-452.
Boniface K, Ducrocq V, Jaubert G, et al. Impact of high-resolution data assimilation of GPS zenith delay on Mediterranean heavy rainfall forecasting[C]//Annales Geophysicae. 2009, 27: 2739-2753.

Q3: line 77: What do you mean with "vertically flat" in this case. Is your statement related to the altitude difference of 344 m? I do not think that this can be considered as "flat".

**Response:** In GNSS tomography, a network whose altitude differences are less than 1 km is regarded as a flat network. Flat networks bring difficulties in retrieving the vertical solutions of the WR. We have clarified this in lines 85-87.

Q4: Line 84: maybe "dry" instead of "rainless".

**Response:** Thank you for your suggestion. And we have used 'dry' instead of 'rainless' in the manuscript.

Q5: Lines 85-96: I think it is a good idea to show the applied parameters for Bernese

in a separate table. In line 92/93, I guess you mean Niell in both cases. Are all GNSS receivers equipped with temperature and pressure measurements? If not, please mention how you derive the ZTD at the receiver locations.

**Response:** According to your and the other reviewer's suggestions, we have moved the description about the GNSS data processing to Appendix A (lines 296-308).
Sorry, it's Niell, we made a typo and has corrected it.
Yes, all the GNSS receivers are equipped with temperature, relative humidity, pressure measurements.

Q6: Lines 99-102: The general purpose of any data assimilation scheme is to obtain the best estimate of the atmosphere not only with respect to ZTD observations. Did you assimilate any other observations than ZTD? It is well known, that one should make use of all available observations to complement each other. Especially as the 3DVAR does not contain any dynamical component. How was the 3DVAR set up? Is it a rapid update cycle with e.g. an hourly update or did you ran the 3DVAR once at the beginning of your period of interest? Did you apply multiple outer loops? What is the ZTD error you used? All these details are important to know as this determines the weight/impact of the observations and the data assimilation in general.

**Response:** We delete the inaccurate expression "In this study, the WRFDA estimates the atmosphere state that best fits the ZTD observations."

The purpose of this manuscript is to conduct an interesting and fair comparison between the tomography technique and the WRFDA. To be fair, we use only GPS data for both tomography and the WRFDA, i.e. slant wet delay for tomography and ZTD for the WRFDA. In addition, except for the GPS data, only the surface meteorological observations can be assimilated into the WRFDA model (the only radiosonde data will be left for validation), but assimilating the surface meteorological data into WRFDA can make very little difference, according to our previous tests.

The physics options are the Kain-Fritsch scheme (Kain and Frisch, 1990), WRF Single-Moment (WSM) 5-class scheme (Hong et al., 2004), unified Noah land-surface model (Tewari et al., 2004), Revised MM5 Monin-Obukhov scheme (Monin and Obukhov, 1954), and Yonsei University planetary boundary layer scheme (Hong et al., 2006). The Rapid Radiative Transfer Model (Mlawer et al., 1997) and Dudhia's scheme (Dudhia, 1989) were used for longwave radiation and shortwave radiation, respectively. (This has been clarified in lines 123-128).

This experiment does not apply multiple outer loops and just run the 3DVAR once at specific epoch such as 0:00 UTC, 21st July in 2015. The ZTD error is output by the Bernese 5.0 software.

Q7: Lines 102-104: Does the model domain only encompass the area shown in Figure 1? If this is the case, you may only have approx. 30*25 grid cells. Assuming a boundary relaxation zone of 5 cells, you effective model domain will be 20*15 cells which is far too small. The model does not have a chance to develop its own state but

is mainly determined by the boundary conditions. Please clarify. Did you apply the default layer settings in WRF by setting "eta_levs" to a certin value or did you define the levels on your own? How many layers are in the PBL? This may be important as the majority of the humidity is located inside the PBL. A lot more information is necessary here.

**Response:** Yes, the model domain only encompasses the area shown in Figure 1 in the manucript. And the relaxing zone is 4 cells. We find that we caused a lot of confusion by running the WRF model. Previously, we first run WRFDA 3DVAR to assimilate ZTDs and generate the output, labeled as output1. Then, we run the WRF using output1 as the background value to generate output2 at the same epoch. Actually, we just want to do data assimilating other than hindcasting, we don't need to run WRF to generate output2. I think running the WRF after 3DVAR caused a lot of confusion, we have removed the WRF part from the manuscript and redone the validation using output1 instead of output2. By these revisions, we have focused on the data assimilation and the WRFDA model. So, the statistic of bias, STD and RMS is same.

We set 46 layers in WRF on our own and 10 layers in the PBL.

In order to figure out how sensitive the wet refractivity output is to the domain size, we carry out a sensitivity test at 00:00 UTC 22$^{nd}$ July in 2015. And we increase the domain size gradually from $30 \times 24$ grids to $190 \times 184$ grids. In each run, we validate the wet refractivity derived by the WRFDA output using the radiosonde data. The statistical results of the sensitivity test are shown in Figure R1. It shows that the smaller domain size has the smaller bias, STD, and RMS. So, the domain size of the data assimilation experiment is set to $30 \times 24$ grids which just cover the study area. This has been discussed in lines 115-138 in the manuscript.

[Figure]

**Figure R1.** Statistics of sensitivity test with different domain size.

Q8: Lines 109-111: Did you apply the reanalysis of ERA-Interim, which has a resolution of 0.75◦ and not 0.125◦ or the operational analysis? Are the forcing data

applied on model levels or on pressure levels? In case you applied the former data, I'm afraid that this is not a suitable data to study the behaviour of a convection permitting model especially at these short time scales although data assimilation is applied.

**Response:** Yes, I applied the reanalysis of ERA-Interim. We use the ERA-Interim data on pressure levels and surface data. Its nominal resolution is $0.125° \times 0.125°$ and the real resolution is $0.75° \times 0.75°$.

Q9: Line 115: To me the applied CV3 method is a major concern. I guess you know that this matrix is derived from a NCEP model climatology at a horizontal resolution of roughly 2∘ and this is applied on the CP scale in your study. I am concerned if this is a scientifically valid approach.

**Response:** ARW version 3 Modeling System User's Guide (published in July 2016) says that "*However, CV3 (a BE file provided with our WRFDA system) is a global BE and can be used for any regional domain, while CV5, CV6, and CV7 BE's are domain-dependent, and so should be generated based on forecast or ensemble data from the same domain.*" in page 6-39.

"*Theoretically, CV3 BE is a generic background error statistics file which can be used for any case.*" in page 6-40.

Based on these, we adopted the default CV3 background error. It may not be the best, but it could be used in this case.

Q10: Line 118: I think the word reanalysis is misleading here as you probably only used ZTD observations. I also do not really see from the Vedel and Huang publication how WR is derived. Please also include the units for k1 and k2. Are T and P only used at the surface? This is not clear here. Also, please use "p" instead of "P" for pressure.

**Response:** Thank you for your suggestion. And we revise the term 'reanalysis' to 'output'. Vedel and Huang (2004) didn't directly give the equation for WR calculation, but we can easily partition the equation for WR calculation from the equation for wet delay calculation. We attach a figure (the code is from da_transform_xtoztd.inc in WRFDA) to show Vedel and Huang's equation as follows. In this figure, wzd is the zenith wet delay, which is calculated by integrating the product of WR and altitude difference (which represented by dh). Therefore, the wet refractivity can be easily derived by dividing the wzd increment by the dh.

The unit of $k_1$ is K/Pa, and the unit of $k_2$ is K²/Pa. We clarified this in the manuscript in line 113. We have replaced "P" with "p".

```
subroutine da_transform_xtoztd(grid)

!-------------------------------------------------------------------
!  Purpose: to compute the Zenith Total Delay, and save it to xb%ztd.
!
!  Both of the wet and dry delay are computed based on Vedel and Huang,
!         J. Meteor. Soc., 82, 459-472, 2004.
!
!        ** Equation (3) in Vedel and Huang is wrong.
!
!               ported by Yong-Run Guo  05/12/2008 from wrf3dvar.
!-------------------------------------------------------------------

   implicit none

   type (domain), intent(inout) :: grid

   integer :: i, j, k

   real    :: const, part, term1, term2, wzd, hzd, zf

   if (trace_use) call da_trace_entry("da_transform_xtoztd")

!--WEIGHTED SUM OF VERTICAL COLUMN      wzd = Σ WR * dh

! Wet delay:                                        dh
      wzd = 0.0
      do k=kts, kte
         const  = (grid%xb%hf(i,j,k+1)-grid%xb%hf(i,j,k)) / a_ew
         part   = grid%xb%p(i,j,k)*grid%xb%q(i,j,k) / grid%xb%t(i,j,k)
         term1  = part * const * wdk1
         term2  = part * const * wdk2 / grid%xb%t(i,j,k)
         wzd    = wzd + term1 + term2
      enddo

! Hydrostatic delay (Saastamoinen 1972):
      zf = (1.0 - zdk2*cos(2.0*grid%xb%lat(i,j)*radian) - zdk3*grid%xb%terr(i,j))
      hzd = zdk1 * grid%xb%psfc(i,j) / zf
```

Q11: Line 137: Do you mean the ECMWF (re-)analysis or the new analysis obtained from WRFDA?

**Response:** The reanalysis means WRFDA output and background data. And the data assimilation output is labeled as Output1. The output from WPS and real.exe is labeled as Output2.

Q12: Line 138: What do you mean with "nearest four grids"?

**Response:** This part has been rewritten, no "nearest four grids" any longer. The new expressions are:

"The vertical coordinates of the Output1 and the Output2 are converted to geopotential heights by NCAR Command Language (NCL) (NCL, 2013) and the geodetic heights of tomographic results are converted to normal height. The slight differences between geopotential heights and normal heights are neglected. We interpolate the Output1 to tomographic nodes since the former has a much higher resolution ~23 layers from 0 to 10 km height than the latter (13 layers) and thus we can get a higher interpolation accuracy. We use a bilinear interpolation method in the horizontal domain and a linear interpolation method in the vertical direction."

Q13: Lines 139-140: Why did you adjust the radiosonde data? This distorts the

radiosonde observations. I strongly recommend to interpolate the GNSS and tomography fields to the radiosonde location. How did you interpolate the unevenly distributed WRF model layers to the tomography layers, which have a constant spacing? The native WRF model output is not on pressure levels but on terrain following coordinates. I think is it necessary to include a short paragraph here.

**Response:** Thank you for your suggestion. In the vertical troposphere, the tomography model only has 13 layers whose vertical resolution is only 800 m while the radiosonde has a vertical resolution of ~23 layers from 0 km to 10 km height. It means the radiosonde data have a much better vertical resolution than the tomography results. Therefore, we think interpolating the dense radiosonde data to the sparse tomography layers in the vertical direction would be more accurate. We show the original radiosonde profiles in Figures 4 and 5 now.

The vertical coordinates of the Output1 and the Output2 are converted to geopotential heights by NCAR Command Language (NCL) (NCL, 2013) and the geodetic heights of tomographic results are converted to normal height. The slight differences between geopotential heights and normal heights are neglected. We interpolate the Output1 to tomographic nodes since the former has a much higher resolution ~23 layers from 0 to 10 km height than the latter (13 layers) and thus we can get a higher interpolation accuracy. We use a bilinear interpolation method in the horizontal domain and a linear interpolation method in the vertical direction. By these methods, we interpolate the WR derived from the Output1, the Output2 and radiosonde data to the tomography nodes. This has been illustrated in lines 182-189 in the manuscripts.

Q14: Line 143: Is "Reanalysis 2" your control run mentioned in line 117 or is this an assimilation run where everything except ZTDs was assimilated?

**Response:** It is the control run. "Reanalysis2" is changed to "Output 2" now which is the output of real.exe. This has been clarified in line 109-110.

Q15: Line 152: Why does this lead to a decrease in performance in the tomography and the WRF model?

**Response:** Figures 4 and 5 show that the WR was distributed evenly from 0 to 10 km in July period when Hong Kong rains heavily, while the WR concentrated from 0 to 6 km in August period. This means that the water vapor varied sharply in the vertical direction in the August period and was relatively smooth in the July period. Both the WRFDA and the tomography technique can better retrieve the water vapor with smooth distribution than that with sharp variations. This has been clarified in lines 196-200.

Q16: Line 154: Did you perform any significance tests or do you mean something like "considerably"?

**Response:** No, we did not. We treat the radiosonde data as the true values and use them to validate the other results. The expression "Compared with Reanalysis2, the Reanalysis1 is slightly improved, but the improvement is not significant" is kind of misleading, we have revised it to Compared with Background output, the WRFDA

output is slightly improved by reducing the mean absolute error (MAE) by 1.25 mm/km.. This has been revised in lines 201-202.

Q17: Line 173: Why does the tomography "may" perform better than WRF? I though you did investigate this?
**Response:** The RMS of the tomography results is smaller than the WRFDA output results at 400 m, 1600 m, and 2400 m height, which is shown in Figure 6f. From the statistical perspective, the tomography performs better than the WRFDA at these heights. We have clarified this and deleted the term "may", see lines 218-219.

Q18: Lines 196-198: This statement is very confusing and queries the results of your study.
**Response:** To be more specific and accurate, we have revised the whole paragraph to (lines 242-246):
"In general, assimilating GNSS ZTD into the WRFDA has slightly improved the WR retrieval by decreasing the RMS by 0.2 mm/km. The WR derived from WRFDA output and Background output has apparently smaller RMS than the tomographic WR (4.15 mm/km vs. 6.50 mm/km and 4.31 mm/km vs. 6.50 mm/km, respectively). The results obtained from WRFDA and tomography are better in the wet period than in the dry period, which is mainly due to the sharp vertical variation of WR in the dry period."

Q19: Line 208: With ZTD you do not assimilate the column water vapour. The signal delay is assimilated from which TCW can be derived.
**Response:** Yes, we just need to assimilate one of them.

Q20: Line 210: From your results I do not agree with this statement.
**Response:** Make full use of the vertical structure information of water vapor could benefit the data assimilation. It could provide more information such as the vertical water vapor distribution than ZTD. And the MAE of assimilating ZTD is 6.04 mm/km while the MAE of assimilating tomographic WR is 5.92 mm/km. This result shows that assimilating tomographic WR could improve the WR retrieve.

Q21: Line 213: How did you assimilate relative humidity only in WRFDA? If you use p and T from reanalysis (again the WRFDA analysis or ECMWF?), the tomography is not model independent anymore.
**Response:** According to the WRF data assimilation scheme, we can only assimilate the relative humidity together with T and p from background data. We tried to mitigate model dependence. We use the T and P from the background data output (O1) without assimilating anything. Then we run WRFDA to assimilate the tomographic WR together with T and p, and then run WRFDA to generate the new output (O2). The T and p in O1 and O2 have very small difference, therefore we controlled the influence of T and p on the final results. The difference between O1 and O2 mainly lies in the humidity, which was caused only by the tomographic WR.

---

## Author Comment (AC7) · 30 Nov 2018

[revised manuscript text omitted]